# *GDF-9* and *BMP-15* mRNA Levels in Canine Cumulus Cells Related to Cumulus Expansion and the Maturation Process

**DOI:** 10.3390/ani10030462

**Published:** 2020-03-10

**Authors:** George Ramirez, Jaime Palomino, Karla Aspee, Monica De los Reyes

**Affiliations:** Laboratory of Animal Reproduction, Department of Animal Production, Faculty of Veterinary Sciences, University of Chile, Santiago 8320000, Chile; GEORGES.RSAB@gmail.com (G.R.); jpalomin@veterinaria.uchile.cl (J.P.); kaspeem.vet@gmail.com (K.A.)

**Keywords:** cumulus expansion, transforming growth factors, transcripts, dog, estrous cycle

## Abstract

**Simple Summary:**

The knowledge of physiological events associated with canine reproduction involving oocyte developmental potential is essential to increase the success of reproductive biotechnologies in this species. In mammals, the oocytes are closely surrounded by a group of cells known as the cumulus cells. Although it is not well-known how these cells interact with the oocyte to promote maturation, they may provide important answers concerning oocyte development. The competence to undergo expansion is a unique characteristic of cumulus cells which is critical for normal oocyte maturation, however, the complete expansion of these cells takes longer in canines, which has been associated with the lengthy maturation process of the oocyte. Growth Differentiation Factor 9 (GDF-9) and Bone Morphogenetic Protein 15 (BMP-15) are described as relevant players in the oocyte–cumulus cells’ regulatory mechanisms. Cumulus cells express many important genes from a very early stage, therefore, we proposed to study the gene expression of *GDF-9* and *BMP-15* in canine cumulus cells in relation to cumulus expansion and the maturation process. We demonstrate, for the first time, that these genes are differentially expressed in canine cumulus cells throughout the estrous cycle and that this expression is related to cumulus expansion and maturity status, suggesting specific regulation.

**Abstract:**

The competence to undergo expansion is a characteristic of cumulus cells (CCs). The aim was to investigate the expression of *GDF-9* and *BMP-15* mRNA in canine cumulus cells in relation to cumulus expansion and meiotic development over the estrous cycle. CCs were recovered from nonmatured and in vitro-matured (IVM) dog cumulus oocyte complexes (COCs), which were obtained from antral follicles at different phases of the estrous cycle. Quantitative real-time polymerase chain reaction (q-PCR) was used to evaluate the relative abundance of *GDF-9* and *BMP-15* transcripts from the CCs with or without signs of expansion. The results were evaluated by ANOVA and logistic regression. The maturity of the oocyte and the expansion process affected the mRNA levels in CCs. There were differences (*p* < 0.05) in *GDF-9* and *BMP-15* gene expression in CCs isolated from nonmatured COCs when comparing the reproductive phases. Lower mRNA levels (*p* < 0.05) were observed in anestrus and proestrus in comparison to those in estrus and diestrus. In contrast, when comparing *GDF-9* mRNA levels in IVM COCs, no differences were found among the phases of the estrous cycle in expanded and nonexpanded CCs (*p* < 0.05). However, the highest (*p* < 0.05) *BMP-15* gene expression in CCs that did not undergo expansion was exhibited in anestrus and the lowest (*p* < 0.05) expression was observed in estrus in expanded CCs. Although the stage of the estrous cycle did not affect the second metaphase (MII )rates, the expanded CCs obtained at estrus coexisted with higher percentages of MII (*p* < 0.05). In conclusion, the differential expression patterns of *GDF-9* and *BMP-15* mRNA transcripts might be related to cumulus expansion and maturation processes, suggesting specific regulation and temporal changes in their expression.

## 1. Introduction

Cumulus cells (CCs), which are closely adjacent to the oocyte, are differentiated cells derived from preantral granulosa cells, the primary cell type in the ovary that provides physical support and the microenvironment required for the developing oocyte. This intimate association allows CCs to play vital roles, such as supporting the maturation of the oocyte and relaying endocrine and paracrine signals. This relationship between the oocyte and the CCs involves a complex and varied set of interactions [1]. The growth of the oocyte and its CC compartment in the follicle takes place in a highly coordinated and mutually dependent manner [2].

During the late stages of preantral follicles, the formation of the follicular antrum begins and separates the granulosa cells into the cumulus and the mural granulosa cells [3]. The mechanisms responsible for this differentiation have not yet been fully described, however, the paracrine factors, such as Growth Differentiation Factor 9 (GDF-9) and Bone Morphogenetic Protein15 (BMP-15), were shown to be involved in promoting the differentiation of preantral granulosa cells into cumulus cells [4,5]. These proteins are members of the Transforming Growth Factor beta (TGF-β) superfamily and are described as relevant players in the oocyte–cumulus cells’ regulatory mechanisms [6], and essential for follicular cell processes and fundamental to oocyte maturation [7,8]. Both GDF-9 and BMP-15 were described to be involved in the communication between oocytes and the adjacent cumulus cells by regulating amino acid uptake, glycolysis, and cholesterol biosynthesis from cumulus cells via gap junctions to the oocyte [5,9]. In addition to the regulation of cumulus cell metabolism, GDF-9 and BMP-15 regulate diverse processes and gene expression during the preovulatory stage [10].

GDF-9 and BMP-15 were identified in canine granulosa/theca cells [11] and in canine oocytes during in vitro maturation (IVM) [12], and their changes during follicular development throughout the dog estrous cycle were also reported [13]. Recent studies using recombinant GDF-9 and BMP-15 in vitro demonstrated that both growth factors in combination also play a significant role in meiotic development in dogs [14].

Cumulus cells communicate with each other and with the oocyte through specialized gap junctions that allow metabolic exchange and the transport of signaling molecules [15]. In mice, during meiotic arrest, cyclic guanosine monophosphate (cGMP) from CCs passes into the oocyte through gap junctions and inhibits the hydrolysis of cyclic adenosine monophosphate (cAMP) by phosphodiesterase (PDE) [16]. This inhibition maintains a high concentration of cAMP in the oocyte and blocks meiotic progression [17]. In most mammals, after the preovulatory peak of luteinizing hormone (LH), cumulus expansion occurs as a critical aspect in the final stages of follicular development [15]. The loss of gap junctions between cumulus cells and the oocyte during cumulus expansion has been closely related to oocyte meiotic progression [1,18], and this process is modulated in part by BMP-15 and GDF-9 [16,19]. Both factors induce a cascade of reactions involving downstream genes [20]. This process initiates the secretion of extracellular matrix materials, such as hyaluronic acid (HA), which is synthesized by the enzyme hyaluronan synthase 2 (HAS2), HA-binding protein pentraxin-3 (PTX3), and tumor necrosis factor-alpha-induced protein 6 (TNFAIP6) [21] (Hussein et al., 2006), thereby enabling the cumulus cells to undergo mucification, which is a critical process in the final stages of follicular development [15].

In canine oocytes, the LH rise is not enough to stimulate meiotic resumption, and although the CCs expand, the innermost layers adjacent to the oocyte remain closely linked to the oocyte for two or three days after ovulation and continue to support the oocyte [22,23]. The role of these paracrine factors in mucification in dogs and the dynamic expression in canine CCs is intriguing. Thus, the aim of this study was to investigate the expression of *GDF-9* and *BMP-15* mRNA in CCs from canine COCs in order to test the hypothesis that the expression of these factors is related to the expansion of CCs and to meiotic development after maturation. 

## 2. Material and Methods 

### 2.1. Animals and Sample Preparation 

Healthy adult (1–6 year) non-pregnant bitches (*n* = 88) undergoing routine ovariohysterectomy at the local Veterinary Hospital were used in this study. All animals were classified considering the stages of estrous cycle and treated according to the guidelines of the Chilean Ethics Committee of the National Commission for Scientific and Technological Research (CONICYT). The owners of the animals gave informed consent for the use of samples.

Serum progesterone (P4) analyses of blood samples obtained on the day of surgery of each donor was used to confirm the physiological status of each bitch. This was assessed by an enzyme-linked fluorescence assay (ELFA) on the Mini-Vidas automated analyzer (Biomerieux, Marcy l’Etoile, France) [24] using P4 canine kits (VIDAS^®^ Progesterone #30409, Biomerieux, Marcy l’Etoile, France).

The reproductive phase was also assessed according to the ovarian structures by examination of the type of growing follicles and corpus luteum (CL) [25].

### 2.2. COCs Processing

Ovaries were kept in saline solution (0.9%, w/v NaCl) containing 100 IU/mL of penicillin G and 0.1 mg/mL of streptomycin sulfate (Merck KGaA, Darmstadt, Germany), then transported to the laboratory within 20 min. During transport, the temperature of the ovaries was maintained at 37 °C. After washing in phosphate-buffered saline (PBS) (137 mM NaCl, 2.7 mM KCL, 10 mM Na_2_HPO_4_, 2 mM KH_2_PO_4_) at 38 °C and pH 7.4, the ovaries were mechanically dissected.

Only antral follicles from 0.5 mm to 4.9 mm (anestrus, proestrus and diestrus) or 5 to 10 mm (estrus) were punctured with a narrow bore Pasteur pipette, and cumulus–oocyte complexes (COCs) with more than three tightly wrapped layers of CCs and uniformly distributed oocyte cytoplasm were selected under a dissecting microscope (Meiji Techno SKT, Tokyo, Japan). The selected COCs retrieved from pooled antral follicles of individual bitches according to each reproductive phase were collected separately in Petri dishes containing PBS. 

### 2.3. Experimental Design

Two experimental groups of COCs were set up according to the experimental design in each replicate (Figure 1).

#### 2.3.1. Noncultured COCs

Groups of COCs at each phase of the estrous cycle (~800–1000 total) were subjected to CC denudation by gentle pipetting under a stereomicroscope, washed twice in 5% PBS (Merck KGaA Darmstadt, Germany), and evaluated on the basis of satisfactory cumulus morphology without signs of mucification. Cumulus cells were stripped off from COCs with a small-bore pipette and transferred into a 1.5 mL conical tube and washed three times in PBS [5]. The denuded oocytes were removed from the dish and were examined for presence of a visible germinal vesicle (GV), as previously described [26].

#### 2.3.2. COCs Subjected to In Vitro Maturation (IVM) 

Cumulus–oocyte complexes (~400) at each phase of the estrous cycle were in vitro-maturated for 72 h in accordance with our previous studies [14]. In brief, COCs were incubated in 100 μL drops of Tissue Culture Medium 199 (TCM199-25 mM Hepes; Earle’s salt, Invitrogen, Carlsbad, CA, USA), with 10% fetal calf serum (FCS) (Merck KGaA), 0.25 mM pyruvate, 10 IU/mL of hCG, 300 IU/ mL of penicillin, 20 μg/mL of streptomycin and, 2 μg/mL of ß-estradiol (E_2_) (E8875-1G. Merck KGaA) for 72 h at 38.5 °C in a humidified atmosphere with 5% CO_2_.

After the IVM period, each COC was examined under an inverted microscope (Nikon TMS 301953 Tokyo, Japan). The cumulus was evaluated and classified on the basis of cumulus expansion as either nonexpanded or expanded. Cumulus cells were mechanically removed from COCs by passage through a narrow glass pipette. The collected CCs that underwent expansion and those that did not expand were washed separately by gentle centrifugation in an Eppendorf 5415D centrifuge (Eppendorf AG, Hamburg; Germany).

The denuded IVM oocytes were washed in PBS solution and fixed in 4% paraformaldehyde in PBS. For meiotic evaluation, the oocytes were incubated with 4,5-diamidino-2phenylindole (DAPI) solution (Thermo Fisher Scientific Inc., Rockford, IL, USA), 1 µg/mL in PBS for 5 min at room temperature (22 °C) and visualized on an inverted epifluorescence microscope (Olympus IX71, Tokyo, Japan). Each oocyte was classified according to the chromatin configuration as a germinal vesicle (GV), meiosis resumption or germinal vesicle breakdown (GVBD), first metaphase (MI), or second metaphase (MII), as previously described [26], considering the phase of the estrous cycle and cumulus expansion. 

### 2.4. Total RNA Extraction and Reverse Transcription Analysis by Real-Time Polymerase Chain Reaction (RT-qPCR)

Cumulus cells from either group of COCs (non-cultured and IVM) were kept in RNAlater (Ambion^®^, Thermo Fisher Scientific Inc.) and subsequently stored at −80 °C until total RNA isolation for further q-PCR analysis.

Total RNA was extracted from the CCs using the column affinity Purification Kit GeneJET^TM^ RNA (Thermo Fisher Scientific Inc.), following the manufacturer’s instructions. The RNA concentration was determined using a Qubit^®^ Fluorometer (Invitrogen, Eugene, OR, USA) with the quantification kit Qubit^®^ RNA Assay (Molecular Probes, Invitrogen). The RNA samples were stored at −80 °C until use. DNA contamination was removed by DNAse I treatment and reverse transcription (RT) was performed using the enzyme conjugate SuperScript^TM^ First-Strand Synthesis System (Invitrogen). The complementary DNA concentration was determined using the Quantification Kit ssDNA Qubit^®^ Assay (Invitrogen, Molecular Probes). All samples were run in duplicates using 10 ng of complementary DNA (cDNA).

Quantitative real-time polymerase chain reaction (RT-qPCR) was used to evaluate the relative abundance of *GDF-9* and *BMP-15* mRNA transcripts in CCs in nonmatured and IVM COCs throughout the estrous cycle. Considering our previous studies, we previously used the Norm Finder algorithm to generate a stability measure, for which a lower value indicates increased stability in gene expression, and used samples taken from different groups to allow a direct estimation of the variation in expression of different genes (*ACTB*, *H2A*, *EEF1B2*, *GAPDH*, *RPS18*, and *PGK1)*. Therefore, it was possible to rank genes using the similarity of their expression profiles. This method estimated intra- and intergroup variation, which were then included when calculating the value of stability of each reference gene. With this information, the gene that was more suitable for normalization was *ACTB* [25]. Canine-specific primers for both genes and β-actin (*ACTB*) as the reference gene were used (Table 1). The results were obtained by the comparative Ct method using ΔΔCt.

### 2.5. Statistical Analysis

A comparison of the relative expression levels of *GDF-9* and *BMP-15* mRNA in expanded and nonexpanded cumulus cells and between the phases of the cycle was performed using ANOVA. The model included the main effect of cumulus expansion, the phase of the estrous cycle, and the interaction between these two factors. Differences among means were identified with Duncan’s tests. The percentage of oocytes in each stage of meiotic maturation in each condition was evaluated by logistic regression. Differences with *p* < 0.05 were considered significantly different.

All analyses were performed using InfoStat Professional Program (Version 2018; National University of Córdoba, Argentina).

## 3. Results

### 3.1. Estrous Cycle Determination

The different phases of the estrous cycle were determined according to P4 levels obtained from blood samples of each bitch. The P4 values were accompanied by the corresponding ovarian examinations in all samples. Thirty bitches showed serum P4 values less than 0.19 ng/mL and an absence of follicles or corpore lutea (CLs) in the ovarian surface, and were therefore considered in anestrus; 12 bitches were in proestrus with P4 values greater than 0.2 ng/mL up to 2 ng/mL and growing small–medium follicles on the surface of the ovaries; 10 were in estrus, showing 2–18 mg/mL P4 and large follicles on the surface of the ovaries; and 26 were at diestrus, with more than 20 ng/mL P4 and with mainly predominant CLs on the ovaries. The intra-assay coefficient of variation (CV) was 15%–23% and the inter-assay CV was 17%–25%.

### 3.2. Expansion of Cumulus Cells

All noncultured COCs had compact CC layers. COCs incubated for IVM showed different patterns of cumulus mucification (Figure 2). Nonexpanded are shown in Figure 2a,b and expanded in Figure 2c,d.

### 3.3. Expression of Cumulus Expansion-Related Transcripts by Cumulus Cells

Real-time polymerase chain reaction was performed in CCs from COCs that were noncultured and from those submitted to IVM in order to evaluate the mRNA transcripts of *GDF-9* and *BMP-15* in dog CCs over the estrous cycle. This evaluation was made considering the maturation process and cumulus expansion.

Both genes were expressed in CCs of COCs originating from all phases of the estrous cycle, whether they were submitted to IVM or not. However, the relative abundance of these genes was different (*p* < 0.05) according to CC expansion and the maturation process (Figure 3). Cumulus cells from noncultured COCs obtained in anestrus and proestrus showed the lowest *GDF-9* mRNA levels (Figure 3a; *p* < 0.05) and the highest was observed in those at estrus (*p* < 0.05). On the contrary, no differences were found between the estrous phases in *GDF-9* mRNA levels in nonexpanded (Figure 3c) or expanded (Figure 3e) CCs from IVM COCs. Regarding *BMP-15* expression, CCs from nonmatured COCs obtained at anestrus and proestrus also showed lower (*p* < 0.05) mRNA levels than those at estrus and diestrus (Figure 3b), whereas CCs from IVM COCs that did not show signs of mucification expressed the highest (*p* < 0.05) *BMP-15* mRNA levels at anestrus (Figure 3d). The *BMP-15* gene expression in expanded CCs (Figure 3f) registered the lowest (*p* < 0.05) value in the estrus phase.

Comparing gene expression between CCs retrieved from non-cultured COCs and those from COCs after IVM, both expanded and nonexpanded (Figure 4), CCs surrounding IVM oocytes obtained in anestrus (Figure 4a,b) and proestrus (Figure 4c,d) expressed significantly higher mRNA levels of *GDF-9* and *BMP-15* than those from nonmatured oocytes. In contrast, gene expression of both genes in CCs collected from IVM COCs during the estrus phase (Figure 4c,f) was lower in IVM COCs (*p* < 0.05) than in those from nonmatured COCs. The majority of COCs after IVM obtained in the estrus phase exhibited almost only expanded CCs, so the number of CCs that did not mucify was too small to extract RNA from. At diestrus (Figure 4g,h), CCs from nonmatured COCs exhibited higher *GDF-9* and *BMP-15* mRNA levels (*p* < 0.05) than nonexpanded CCs after IVM. However, COCs with cumulus expansion after IVM expressed higher (*p* < 0.05) mRNA levels compared to nonmatured COCs.

### 3.4. Meiotic Growth and Cumulus Expansion 

All oocytes from nonmature COCs were at the GV stage without signs of mucification. The oocyte meiotic maturity status after IVM according to CC expansion was recorded for each phase of the estrous cycle (Table 2). The percentage of oocytes that remained in GV class was similar to expanded and nonexpanded CCs after IVM in estrus and diestrus, but no oocytes at the GV stage were observed from COCs with expanded CCs in anestrus and proestrus. The resumption of meiosis (GVBD) at proestrus and diestrus was higher (*p* < 0.05) in oocytes with expanded CCs in comparison to those that did not expand. In the estrus phase, only oocytes with expanded CCs reached the MII stage.

## 4. Discussion

The expression of the cumulus expansion-related transcripts *GDF-9* and *BMP-15* was studied in canine CCs using nonmatured COCs and IVM COCs. The gene expression of these paracrine factors found in CCs in this work was previously reported in canine ovaries [25,27], as in other animal species [28,29]. The expression levels of these transcripts in CCs were stage dependent over the estrous cycle, which was in agreement with our previous studies performed in canine oocytes and granulosa/theca cells ex vivo [25].

The Figure 3 shows the major mRNA levels of both genes observed in noncultured COCs at estrus and diestrus compared to those at anestrus and proestrus, which may have been influenced by the rise in LH. Cumulus–oocyte complexes obtained at estrus and diestrus were retrieved from follicles that were already exposed to the LH surge in vivo, in contrast to those obtained in the other phases. In dogs, anestrus and proestrus precede the LH peak [30], with the LH surge being a critical event in the CCs development in many species [31], including canines [23]. It was reported that human chorionic gonadotropin (hCG, with LH-like activity) supplementation more effectively promoted CC development at 24 and 36 h of IVM of silver fox oocytes [32]. Similarly, the addition of hCG to culture medium had a highly significant effect on cumulus and oocyte maturation in dogs [33]. In mice, the transcriptome and function of CCs is highly affected by the LH surge [34]. Gonadotropin signal transduction by the somatic compartment requires de novo mRNA synthesis within CCs [35]. Indeed, the LH peak is believed to induce multiple intracellular signaling and second messengers in the follicular cells of several species [36], thereby influencing gene expression. 

Interestingly, the dynamic expression of *GDF-9* and *BMP-15* genes in CCs changed according to the maturity process (IVM vs. non-cultured,) and mucification patterns in the different phases of the estrous cycle, as indicated in Figure 4. Cumulus cells obtained from COCs during anestrus and proestrus submitted to IVM increased the relative abundance of *GDF-9* and *BMP-15* mRNA during culture, especially in CCs that could expand. The culture medium was supplemented with hCG and estrogen, which enhanced CC development, as previously described in dogs [33,37]. This increase in gene expression in IVM COCs was coincident with other reports in humans [38] that describe *GDF-9* and *BMP-15* mRNA to be significantly correlated with the maturation process. However, it was found that CCs from canine COCs retrieved at estrus showed lower levels of *GDF-9* and *BMP-15* mRNA in expanded CCs post-IVM compared with nonmatured COCs. A decrease in the relative abundance of transcripts was suggested as the major change occurring during the transition to the large antral stage of development, which coincides with the acquisition of oocyte developmental competence [39,40]. This finding is supported by other studies that showed reduced *GDF-9* gene expression in human CCs of matured oocytes [41]. Possibly, the reduced transcriptional activity observed in other species in COCs from preovulatory follicles is achieved only after in vitro maturation in canines. In the present study, all COCs obtained in the estrus phase came from large or preovulatory follicles, therefore, the COCs from these matured follicles may have been able to accumulate a large number of transcripts during their growth, resulting in diminished transcriptional activity during the final stages of development in vitro, which possibly occurs in vivo in canines post-ovulation. Accordingly, the protein expression level of GDF-9 in dog oocytes and CCs declined during in vitro maturation and was not elevated at the time of cumulus expansion [12]. 

Cumulus cells obtained from COCs during diestrus and submitted to IVM increased (*GDF-9*) or maintained (*BMP-15*) the transcript levels in expanded CCs only compared to those from noncultured COCs. This suggested that the COCs with low levels of *GDF-9* and *BMP-15* gene expression in their CCs during diestrus were not able to undergo expansion. The presence of a corpus luteum (CL) and/or progesterone in mice affects estradiol concentration, which is required for the acquisition and maintenance of CCs to undergo expansion [42]. In bovines, the presence of a CL in the ovary exerts a negative effect on the expansion of CCs, suggesting that the CL receives the highest rate of blood flow out of all the other ovarian tissues [43].

Clearly, there was a relationship between CC expansion and the relative gene expression of *GDF-9* and *BMP-15*, which changed according to the reproductive status. Therefore, the ability to express *GDF-9* and *BMP-15* genes in the CCs at different stages seemed to be directly influenced by the previous follicular environment, which changes in every reproductive phase.

The effect of the estrous cycle on in vitro meiotic development of canine oocytes has been described in several reports, however, few studies have evaluated the possible effects of the estrous cycle on CC expansion and gene expression related to meiotic growth. The competence to undergo expansion is a unique characteristic of CC differentiation [44], which is critical for normal oocyte maturation in many mammals [45]. Transcriptome analysis in CCs in other species indicated that CC gene expression correlates with oocyte developmental potential [46]. However, in canines, it was suggested that oocytes with large expanded CCs after IVM do not necessarily achieve meiosis [47]. In the present study, we found a clear difference in the expression of the *GDF-9* and *BMP-15* genes in CCs according to the phases of the cycle, however, we did not find such differences in meiotic maturation rates (see Table 2). Meiotic development in oocytes from follicles at different estrous phases was low and within the range reported for in vitro-matured canine oocytes [12,14,27,47]. Although the percentages of MII oocytes were not different among the reproductive phases, the most remarkable result of the meiotic development analyses was the relationship of the cumulus expansion with the percentage of oocytes at the MII stage, but only during the estrus phase. At the same time, there was a significantly lower expression of both genes in expanded CCs from IVM COCs retrieved at estrus in comparison to non-cultured COCs, suggesting that other factors possibly play a role in the meiotic stimulus at the final stage of development. However, these proteins seem to be important in acquiring meiotic competence, because the addition of recombinant GDF-9 and BMP-15 in oocyte maturation media was shown to enhance the competence of canine oocytes matured in vitro [14]. This effect is possible achieved in the early development stages. The process of competence, which includes coordinated sequences of molecular events required for the oocyte to fulfill maturation, is poorly understood in canines. Thus, the dynamics of these signals in regulating meiotic maturation in growing oocytes implies that the oocytes pass through different endocrine and paracrine environments that influence the subsequent expression of different genes. The reproductive cycle in canines is quite long, therefore, the growing oocytes need to pass slowly under these different environments inside the follicle in order to acquire the necessary competence to resume meiosis. Consequently, the fate of individual transcripts stored in the CCs can differ during development. 

## 5. Conclusions

Although we did not evaluate the gene expression in CCs from COCs at each meiotic stage separately, the observed differential expression patterns of *GDF-9* and *BMP-15* mRNA transcripts in CCs during the different estrous phases suggest a possible relationship with cumulus expansion and maturity condition, which could indicate specific regulation and temporal changes in their expression, indicative of different stimulation at precise stages of development.

## Figures and Tables

**Figure 1 animals-10-00462-f001:**
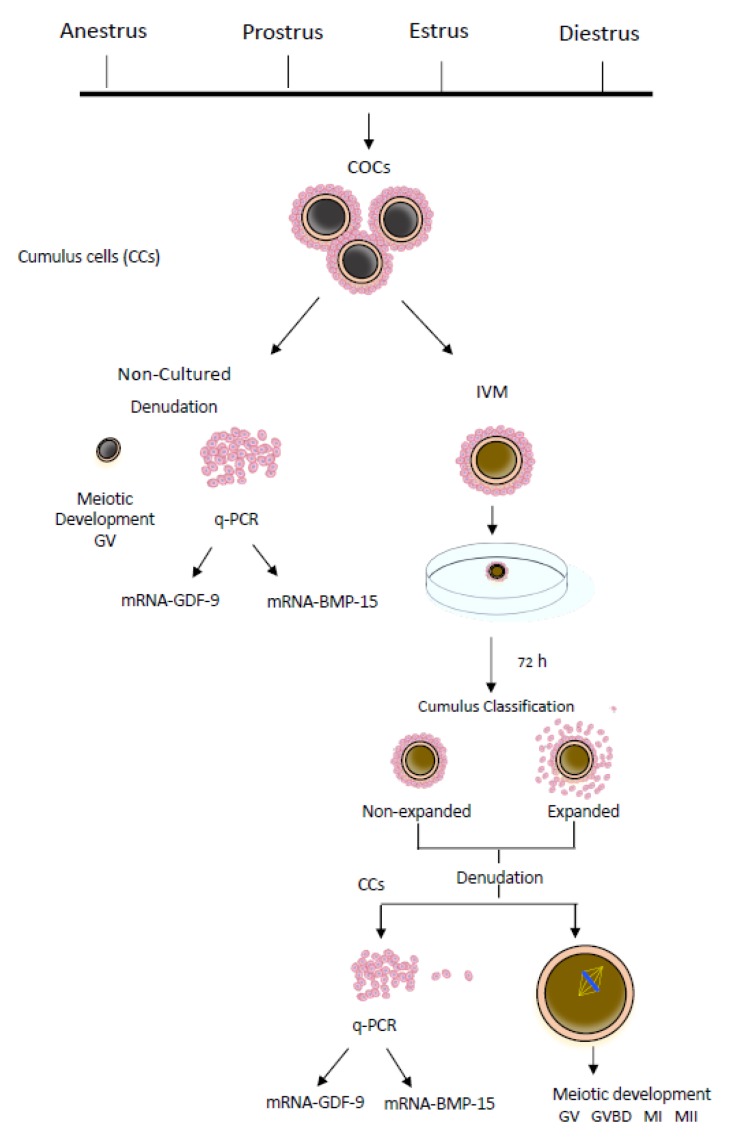
Experimental design. CCs: cumulus cells; COCs: cumulus–oocyte complexes; IVM: in vitro maturation; q-PCR: quantitative real-time polymerase chain reaction; GV: germinal vesicle; GVBD: resumption of meiosis or germinal vesicle breakdown; MI: first metaphase; MII: second metaphase.

**Figure 2 animals-10-00462-f002:**
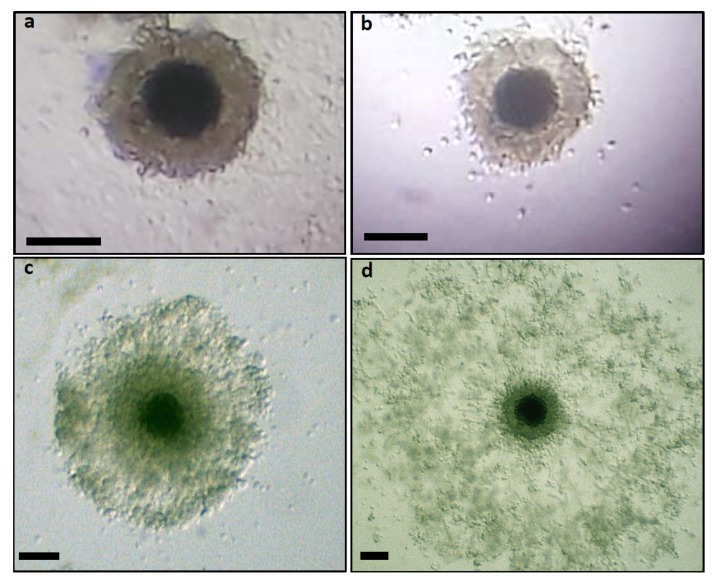
Canine cumulus–oocyte complexes (COCs) after 72 h of in vitro maturation. (**a**,**b**) COCs without cumulus cell expansion; (**c**,**d**) COCs with different grades of cumulus cell expansion (bar: 100 μm).

**Figure 3 animals-10-00462-f003:**
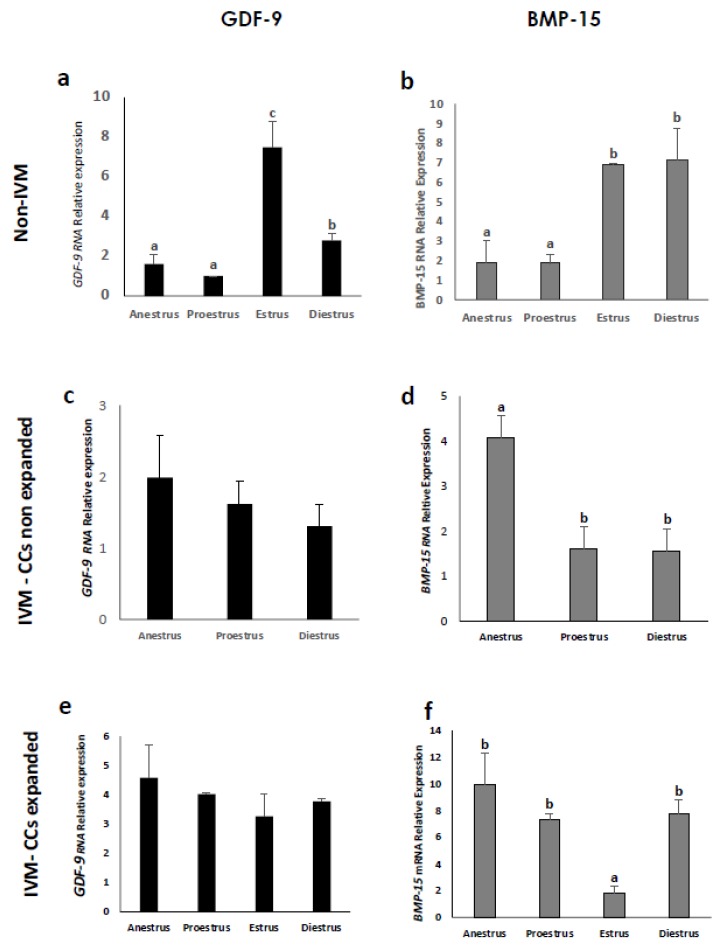
Relative gene expression of Growth Differentiation Factor 9 (*GDF-9*) and Bone Morphogenetic Protein 15 (*BMP-15*) from cumulus cells of nonmatured COCs (**a**,**b**) (~350 COCs at each phase of the estrous cycle), in vitro-matured non-expanded COCs (**c**,**d**) (~300 COCs at anestrus, proestrus and diestrus), and in vitro-matured expanded cumulus cells (**e**,**f**) (~300 at each phase). The mRNA levels were expressed in relation to β-Actin mRNA as the control or housekeeping gene. Different letters above the bars indicate differences at *p* < 0.05.

**Figure 4 animals-10-00462-f004:**
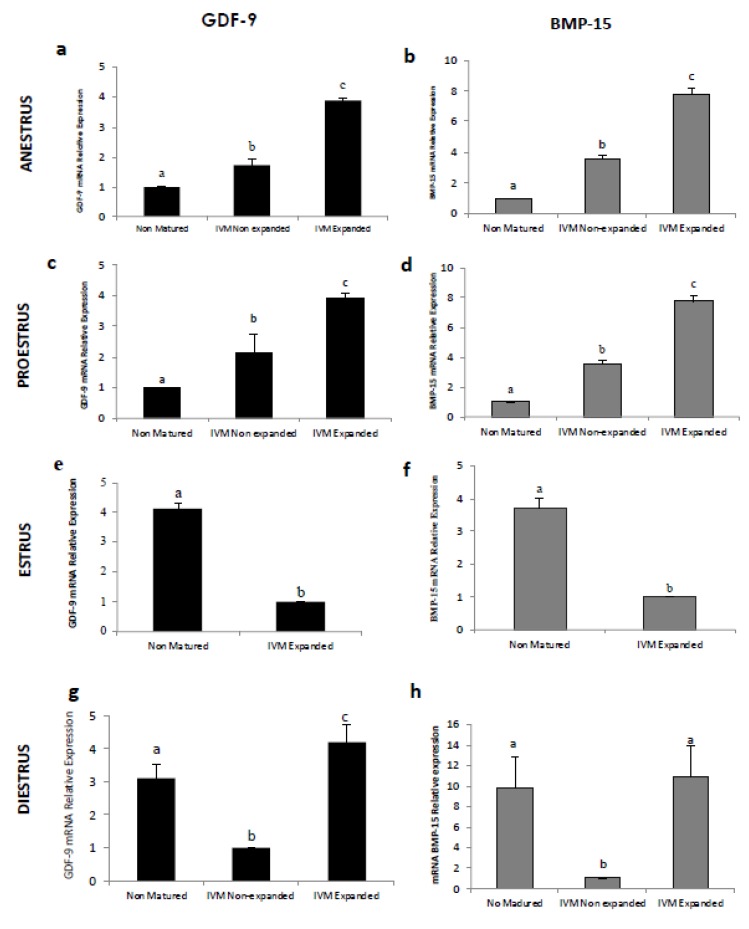
Relative gene expression of Growth Differentiation Factor 9 (*GDF-9*) and Bone Morphogenetic Protein 15 (*BMP-15*) in canine cumulus cells according to the maturation and expansion of the cumulus cells at anestrus (**a**,**b**), proestrus (**c**,**d**), estrus (**e**,**f**), and diestrus (**g**,**h**). The *GDF-9* and *BMP-15* mRNA levels were expressed in relation to β-Actin mRNA as the control or housekeeping gene. For each gene (*GDF-9* or *BMP-15*), bars with different superscripts show a significant difference at *p* < 0.05.

**Table 1 animals-10-00462-t001:** Sequences of GDF-9 and BMP-15-specific primers and reference gene used in this study for qRT-PCR analysis.

Gene	Oligos (5’–3’)	Amplicon (bp)	Accession Number	References
*ACTB*	F: ATTGTCATGGACTCTGGGGATG	191	AF021873.2	[25]
	R: TCCTTGATGTCACGCACGAT			
*GDF-9*	F: TACCCCCATCCCTGCTTTTA	155	NM001168013.1	[27]
	R: TCCACCTTCAGTCGATTCCT			
*BMP-15*	F: CCCTGCCCCTGATTCGGGAG	82	XM003640274.3	[25]
	R: CCGCAAAGGATGCCCAAGGAC			

qPCR: quantitative real-time polymerase chain reaction.

**Table 2 animals-10-00462-t002:** In vitro canine meiotic development according to the expansion of cumulus cells and the estrous cycle.

Oocytes Meiotic Stage (%)						
Cycle	Exp	GV	GVBD	MI	MII	N
Anestrus	+	0 ^a^	37 ^a,b^	53 ^a,b^	10 ^a,b^	51
	−	7 ^b^	37 ^a,b^	48 ^a,b^	7 ^a^	54
Proestrus	+	0 ^a^	60 ^c^	36 ^a^	4 ^a^	53
	−	9 ^b^	21 ^a^	57 ^a,b^	13 ^a,b^	56
Estrus	+	6 ^b^	21 ^a^	56 ^a,b^	18 ^b^	34
	−	5 ^b^	26 ^a^	68 ^b^	0 ^a^	19
Diestrus	+	4 ^b^	50 ^b,c^	39 ^a^	7 ^a^	46
	−	5 ^b^	22 ^a^	56 ^a,b^	17 ^b^	59

GV: germinal vesicle; GVBD: germinal vesicle breakdown; MI: first metaphase; MII: second metaphase; N: number of evaluable oocytes. Expansion: (+) cumulus cells in expansion; (−) nonexpanded cumulus cells. COCs were cultured for 72 h; ^a,b,c^: Within a column, values with different superscript letters differ significantly (*p* < 0.05).

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
