# Peer review of "GDF-9 and BMP-15 mRNA Levels in Canine Cumulus Cells Related to Cumulus Expansion and the Maturation Process"

_animals, 2020, doi:10.3390/ani10030462_

Round 1

Reviewer 1 Report

The authors have revised the manuscript accordingly.  This is a very nice presentation.  I have not further comments.

Author Response

The authors have revised the manuscript accordingly.  This is a very nice presentation.  I have not further comments.

Thank you very much 

Reviewer 2 Report

MDPIanimals-707999R1

The selection of housekeeping gene seems to be an important procedure in ensuring the quality of the experiment. Therefore, it would be better to describe the selection procedure with citation of related reference/s in Materials and Methods.

In Discussion, when the information and results obtained from this experiment are described, the appropriate Tables and Figures should be cited.

Author Response

Point 1:The selection of housekeeping gene seems to be an important procedure in ensuring the quality of the experiment. Therefore, it would be better to describe the selection procedure with citation of related reference/s in Materials and Methods.

Response 1: It was described now

Point 2: In Discussion, when the information and results obtained from this experiment are described, the appropriate Tables and Figures should be cited.

Response 2: now they are cited

This manuscript is a resubmission of an earlier submission. The following is a list of the peer review reports and author responses from that submission.

Round 1

Reviewer 1 Report

Materials and Methods

L148: Describe how ACTB was selected as an internal control gene among a number of housekeeping genes.

Results

L207: Fig. 3e might be Fig.3d.

L212-213: Which gene?

Discussion

In discussion, please explain with reference/citation to Tables and Figures in this study.

Table 2

Indicate incubation time in title or footnote.

Fig. 2 shows that there is a significant difference in the degree of cumulus expansion after 72 hours in culture. Therefore, please indicate the relationship between the meiotic stage and the difference in the degree of cumulus expansion.

Please discuss results of MII rates compared to other previous reports in Discussion.

Is there a lack of accuracy in the comparison between experimental groups with lower developmental rates of MII due to the small number of oocytes examined?

Figure 3

Please indicate the number of samples examined in each experimental group.

In addition, IVM-non expanded had no estrus group. I do not think the experiment was completed.

It may also be necessary to compare and discuss the relationship between non-IVM, IVM-non expanded and IVM-expanded.

Is it possible to combine Fig.3 and Fig. 4 and show it?

Figure 4

Please indicate the number of samples examined in each experimental group.

The values of the expression levels in the same experimental group are different between Fig. 3 and Fig. 4. Why is that?

Author Response

Response to Reviewer 1 Comments

We sincerely appreciate the comments that significantly improved our work.

Point 1: Materials and Methods

L148: Describe how ACTB was selected as an internal control gene among a number of housekeeping genes.

Response 1:  According to our previous study (Palomino J, De los Reyes M. 2016. Temporal expression of GDF-9 and BMP-15 mRNAs in canine ovarian follicles. Theriogenology 86:1541-1549, 2016. http://dx.doi.org/10.1016/j.theriogenology.2016.05.013 ), the expression stability of ACTB, H2A, EEF1B2, GAPDH, RPS18, and PGK1 genes was assessed among canine follicular cell samples. We used the Norm Finder algorithm which generates a stability measure for which a lower value indicates increased stability in gene expression, and we used samples taken from different groups to allow direct estimation of the variation in expression of different genes. Therefore, it was possible to rank genes using the similarity of their expression profiles. This method estimates the intra and intergroup variation, which are then included when calculating the value of stability of each reference gene. With this information, the gene that was more suitable for normalization was selected and this was ACTB. 

Point 2: Results

L207: Fig. 3e might be Fig.3d

Response 2: Yes, you are right, thank you for noticing. It was a mistake and was changed

Point 3: Results

L212-213: Which gene?

Response 3: Both genes. It was corrected now

Point 4: Discussion

In the discussion, please explain with reference/citation to Tables and Figures in this study.

Response 4: I am afraid, but I don’t understand well, do you mean to refer the number of tables or figures in the Discussion section? because the figures and tables are already cited in the Results Section. In the Discussion section there are many references (papers) cited, please let me know if some is missing

Point 5: Table 2

Indicate incubation time in title or footnote

Response 5: Now It was indicated according to your suggestion

Point 6: Fig 2

Fig. 2 shows that there is a significant difference in the degree of cumulus expansion after 72 hours in culture. Therefore, please indicate the relationship between the meiotic stage and the difference in the degree of cumulus expansion.

Response  6: We differentiate in expanded and non-expanded. We did not classify in the degree of expansion because there would have been many variables involved. In Figure 2 we only show what was considered as expanded and what was not considered as expanded. In Table 2, we indicate the relationship between meiotic development according to the expansion of cumulus cells in every phase of the estrous cycle. In the Discussion section we extend the discussion in those points according to your suggestion

Point 7:

Please discuss the results of MII rates compared to other previous reports in Discussion.

Response  7: It was discussed now

Point 8:

Is there a lack of accuracy in the comparison between experimental groups with lower developmental rates of MII due to the small number of oocytes examined?

Response  8: We understand that because it will always be better to have more samples for a better analysis, but more than 70 COCs throughout different replicates were used to meiosis development in every estrous phase. In estrus most of the COCs expanded their cumulus cells, so we had to collect more COCs in order to get more non-expanded COCs, but it was very difficult because in that phase most of them expand their CCs 

Point 9: Figure 3

Please indicate the number of samples examined in each experimental group.

Response  9: It was indicated according to your suggestion

Point 10: In addition, IVM non-expanded had no estrus group. I do not think the experiment was completed.

Response  10: It was contemplated in the experiment, but it was not possible, because as we mentioned before, it was not possible to obtain many COCs in estrus with their cumulus cells without expansion, the vast majority expanded their cumulus cells in this phase, so the amount of CCs without signs of mucification was too small to extract RNA. In any case, the results are interesting despite not having this data. Now we point it in the results section to make it clearer why that group is missing

Point 11: It may also be necessary to compare and discuss the relationship between non-IVM, IVM non-expanded and IVM-expanded.

Response  11: We agree. We extended the discussion in those points now but trying to summarize the major finding. The gene expression of both factors changed in the different phases of the estrous cycle according to these conditions, but the MII rates did not change much among different conditions 

Point 12: Is it possible to combine Fig.3 and Fig. 4 and show it?

Response  12: We believe that it is easier to understand in two separate figures in order to distinguish the statistic differences between the gene expression according to the state of expansion in cumulus cells in each phase and according to the phases at the same state of expansion

Point 13: Figure 4

Please indicate the number of samples examined in each experimental group.

Response  13: A: It was indicated now according to your suggestion

Point 14: The values of the expression levels in the same experimental group are different between Fig. 3 and Fig. 4. Why is that?

Response  14: Because data used for the relative expression analysis in each figure are the threshold cycle (Ct) obtained for both, target and housekeeping (reference) genes, in each experimental condition. Although these Ct values are the same, the relative expression value depends on the experimental conditions compared in each case. Figure 3 represents the relative expression analysis between non-culture and in vitro matured expanded and non-expanded. And in figure 4 it is compared between stages of the estrous cycle.

Reviewer 2 Report

General comments: The study examined GDF-9 and BMP-15 mRNA expression in dog cumulus cells obtained during various reproductive cycles and compared between expand versus non-expand cumulus cells after 72 h IVM. The authors reported that both GDF-9 and BMP-15 expression was lowest in cumulus cells obtained during anestrous and proestrous periods in non-cultured COCs. After IVM, while GDF-9 expression did not differ across reproductive stages and between expanded versus non-expanded cumulus cells, BMP-15 level was higher in expanded cells during estrus and lowest in anestrus donors. Some oocytes (4-18%) achieved MII in all reproductive stages regardless of the status cumulus expansion, except that none of non-expanded cumulus cells in estrus donor completed meiotic maturation. Based on these findings, it seems that GDF-9 and BMP-15 expression is linked to reproductive stage and the status of cumulus expansion. However, the study did not demonstrate the relationship between gene expression and oocyte maturation as claimed by the authors (in the abstract and discussion). To establish such link, cumulus cells from oocytes at each meiotic stage must be assessed separately.  Since this was not done in the present study, the authors should remove the statement that GDF-9 and BMP-15 expression was link to maturity status from the manuscript.

Specific comments:

Line 142: Please provide the method used for removing cumulus cells

Results: None of the figures was included in the manuscript file.

Author Response

Response to Reviewer 2 Comments

We sincerely appreciate your comments that significantly improved our work.

Point 1:

General comments: The study examined GDF-9 and BMP-15 mRNA expression in dog cumulus cells obtained during various reproductive cycles and compared between expand versus non-expand cumulus cells after 72 h IVM. The authors reported that both GDF-9 and BMP-15 expression was lowest in cumulus cells obtained during anestrous and proestrus periods in non-cultured COCs. After IVM, while GDF-9 expression did not differ across reproductive stages and between expanded versus non-expanded cumulus cells, BMP-15 level was higher in expanded cells during estrus and lowest in anestrus donors. Some oocytes (4-18%) achieved MII in all reproductive stages regardless of the status cumulus expansion, except that none of the non-expanded cumulus cells in estrus donor completed meiotic maturation. Based on these findings, it seems that GDF-9 and BMP-15 expression is linked to the reproductive stage and the status of cumulus expansion. However, the study did not demonstrate the relationship between gene expression and oocyte maturation as claimed by the authors (in the abstract and discussion). To establish such a link, cumulus cells from oocytes at each meiotic stage must be assessed separately.  Since this was not done in the present study, the authors should remove the statement that GDF-9 and BMP-15 expression was link to maturity status from the manuscript.

Response 1:  You are right, we cannot conclude a direct relationship between the degree of meiotic maturation and expression of GDF-9 and BMP-15. But, considering that the expression of GDF-9 and BMP-15 genes in CCs changed according to the maturity process (cultured COCs vs non-cultured COCs) and mucification in the different phases of the estrous cycle, we just suggest a possible relationship in this new version of the manuscript

Specific comments:

Point 2:

Line 142: Please provide the method used for removing cumulus cells

Response 2: It was provided now according to your suggestion

Point 3: Results: None of the figures was included in the manuscript file.

Response 3: We regret that, but we uploaded all the figures to the system in a separate file. We don’t know what happened
